# Breaking the Moments Condition Barrier: No-Regret Algorithm for Bandits with Super Heavy-Tailed Payoffs

**Han Zhong**
Center for Data Science, Peking University
hanzhong@stu.pku.edu.cn

**Jiayi Huang**
Center for Data Science, Peking University
Pazhou Lab
jyhuang@stu.pku.edu.cn

**Lin F. Yang** [*]
Department of Electrical and Computer Engineering,
University of California, Los Angles
linyang@ee.ucla.edu

**Liwei Wang** [*]
Key Laboratory of Machine Perception,
MOE, School of EECS,
Institute for Artificial Intelligence,
Peking University
wanglw@cis.pku.edu.cn

## Abstract

Despite a large amount of effort in dealing with heavy-tailed error in machine learning, little is known when moments of the error can become non-existential: the random noise $\eta$ satisfies $\Pr[|\eta| > |y|] \leq 1/|y|^{\alpha}$ for some $\alpha > 0$. We make the first attempt to actively handle such super heavy-tailed noise in bandit learning problems: We propose a novel robust statistical estimator, mean of medians, which estimates a random variable by computing the empirical mean of a sequence of empirical medians. We then present a generic reductionist algorithmic framework for solving bandit learning problems (including multi-armed and linear bandit problem): the mean of medians estimator can be applied to nearly any bandit learning algorithm as a black-box filtering for its reward signals and obtain similar regret bound as if the reward is sub-Gaussian. We show that the regret bound is near-optimal even with very heavy-tailed noise. We also empirically demonstrate the effectiveness of the proposed algorithm, which further corroborates our theoretical results.

## 1 Introduction

Multi-armed bandit (MAB) problems have been introduced by Robbins (Robbins, 1952), and have since become a standard model for modeling sequential decision-making problems. In an MAB instance, there are finite number of arms, pulling each of which an agent receives a random reward (payoff) with unknown distribution. An agent then aims to maximize the received rewards by pulling the arms strategically for a number of times. The MAB problems frequently arise in practice: e.g., clinical trials and online advertising. When the number of arms becomes infinite in practice, the stochastic linear bandit (Abe and Long, 1999; Auer, 2002; Dani et al., 2008) generalizes the classical MAB by assuming the underlying reward distribution possesses a linear structure. Linear bandits achieve tremendous success such as online advertisement and recommendation systems (Li et al., 2010; Chu et al., 2011; Li et al., 2016).

---

[*]Corresponding author.

35th Conference on Neural Information Processing Systems (NeurIPS 2021).

Due to the online learning nature of a bandit problem, we measure the performance of an agent via regret, which measures the differences of the rewards collected from the best arm to those collected from the agent. When the reward distribution is benign, e.g., with sub-Gaussian tails[†], there are a number of efficient algorithms (Bubeck and Cesa-Bianchi, 2012; Lattimore and Szepesvári, 2020), which obtain a worst-case regret bound of the form $\widetilde{O}(\sqrt{AT})$[‡], where $A$ is the number of arms and $T$ is the total number of arm pulls. Any algorithm with regret bound sublinear in $T$ is effectively learning as its average regret tends to 0 when $T \to \infty$. In the linear setting, a number of results (Dani et al., 2008; Abbasi-Yadkori et al., 2011) achieve $\widetilde{O}(\text{poly}(d)\sqrt{T})$ regret bounds, eliminating the dependence on $A$. Here $d$ is the ambient dimension of the problem. A related performance measure is about the number of pulls to identify the best arm (Soare et al., 2014; Tao et al., 2018; Jedra and Proutiere, 2020). We note that usually a regret minimization algorithm can be converted to identify the best arm with high probability.

Nevertheless, in many practical scenarios, we encounter non-sub-Gaussian noises in the observed payoffs, e.g., the price fluctuations in financial markets (Cont and Bouchaud, 2000; Rachev, 2003; Hull, 2012), and/or fluctuations of neural oscillations (Roberts et al., 2015). In such scenarios, the previously mentioned algorithms may fail. To tackle this problem, Bubeck et al. (2013) makes the first attempt to study the stochastic MAB problem with heavy-tailed noise. Specifically, for the rewards with a finite second order moment, by utilizing more robust statistical estimators, Bubeck et al. (2013) achieves regret bound of the same order as in the bounded/sub-Gaussian loss setting. Then, Medina and Yang (2016); Shao et al. (2018); Xue et al. (2020) study the heavy-tailed linear bandits. They consider a general characterization of heavy-tailed payoffs in bandits, where the reward takes the form $r = \mu + \eta$, where $\mu$ is an unknown but fixed number and $\eta$ is a random noise, whose distribution has a finite moment of order $1 + \epsilon$. Here $\epsilon \in (0, 1]$. For this setting, they establish a sublinear regret bound $\widetilde{O}(T^{\frac{1}{1+\epsilon}})$. Unfortunately, when $\epsilon = 0$, these regret bounds will be linear in $T$, failing to learn in such situations. To further account for many such real-world scenarios, where the payoff noise has super-heavy tails (e.g., only $(1 + \epsilon)$-th moment for $\epsilon \in (0, 1)$ exists, or Cauchy distribution whose mean does not exist), new algorithms need to be developed:

*Can we design an efficient algorithm that provably learns for bandits with super heavy-tailed payoffs?*

In this paper, we give the affirmative answer to this question. Without loss of generality, we consider the linear setting, which includes MAB as a special case. In this setting, each arm is viewed as a vector in $\mathbb{R}^d$. The random reward of the arm $x$ is specified as $\theta^\top x + \eta$, where $\theta \in \mathbb{R}^d$ is an unknown but fixed vector and $\eta$ is a super heavy-tailed symmetric random noise such that $\Pr(|\eta| > y) \le 1/y^\alpha$ for any $y > 0$ and some $\alpha > 0$. One of the key challenges in this setting is that the mean of $y$ may not exist. The previous robust mean estimators (Bubeck et al., 2013), such as truncated empirical mean and median of means (Bubeck et al., 2013; Medina and Yang, 2016; Shao et al., 2018; Xue et al., 2020) which require the estimation of the mean, cannot effectively handle this super heavy-tailed noise. On the other hand, since the mean does not exist, we are also required to measure the performance of the agent with high-probability pseudo-regret (defined in Section 2.2). To tackle these challenges, we propose a novel estimator: mean of medians. We then present a generic algorithmic framework to apply it in any existing bandit algorithm. Below, we summarize our contributions:

- We propose a novel robust statistical estimator: mean of medians. Specifically, we simply split $\widetilde{n}$ samples into $k$ blocks and takes the mean of the median in each block. Theoretically, we can prove that, by utilizing $\widetilde{n}(\alpha)$ samples, the super heavy-tailed noise is reduced to the bounded noise with high probability. Here $\widetilde{n}(\alpha)$ is a constant which depends on $\alpha$.

- For the super heavy-tailed linear bandits, we propose a new algorithmic framework. In detail, by simply combing the above mean of medians estimator and an arbitrary provably efficient bandit algorithm, we obtain a new algorithm that can be proved efficient for regret minimization problems and best arm identification problems. Our obtained sample bounds and regret bounds can be nearly optimal.

---

[†]For any $\zeta > 0$, a random variable $X$ is said to be $\zeta$-sub-Gaussian if it holds that $\mathbb{E}[e^{t(X - \mathbb{E}[X])}] \le e^{\zeta^2 t^2/2}$ for any $t > 0$.

[‡]$\widetilde{O}(\cdot)$ ignores logarithm factors.

- We instantiate our framework with Student's $t$-noises and compare with previous methods. Our experiments demonstrate our method can significantly outperform existing algorithms in these environments, and is strictly consistent with our theoretical guarantees.

## 1.1 Related Works

A line of recent work (Bubeck et al., 2013; Medina and Yang, 2016; Shao et al., 2018; Xue et al., 2020) on the heavy-tailed MAB or linear bandits uses the truncated empirical mean and median of means as the robust estimators. However, without assuming the finite moments of order $1 + \epsilon$ for some $\epsilon \in (0, 1]$, it remains unclear whether one can attain equivalent regret/sample complexity for the more heavy-tailed setting, e.g, the noise of the payoff follows a Student's $t$-distribution, or whether sublinear regret algorithms of any form are even possible at all. In comparison, by incorporating the mean of medians estimator, we can design a general provably efficient algorithmic framework for the super heavy-tailed linear bandits.

Our work also adds to the vast body of existing literature on the regret minimization problem on linear bandits (Auer, 2002; Dani et al., 2008; Rusmevichientong and Tsitsiklis, 2010; Abbasi-Yadkori et al., 2011; Lattimore and Szepesvari, 2017; Combes et al., 2017). A remarkable analysis is given by Auer (2002) who builds confidence regions for the true model parameter and then optimistically selects the action minimizing the loss over these sets. In the theoretical view, for the setting where the arm set is finite, Auer (2002) establishes a $\widetilde{O}(\sqrt{dT})$ regret bound. Then, Dani et al. (2008); Rusmevichientong and Tsitsiklis (2010); Abbasi-Yadkori et al. (2011) construct the confidence ellipsoids for the setting where the arm set is infinite and establish a $\widetilde{O}(d\sqrt{T})$ regret bound.

Our work is also closely related to another line of work (Soare et al., 2014; Soare, 2015; Garivier and Kaufmann, 2016; Tao et al., 2018; Xu et al., 2018; Fiez et al., 2019; Jedra and Proutiere, 2020) on the best arm identification problem for linear bandits. This problem is also referred as pure exploration problem since there is no price to be paid for exploring and thus we don't need to carefully balance exploration against exploitation. Hence, an algorithm that is optimal for the best arm identification problem may be suboptimal for the regret minimization since its exploration is too aggressive. The reverse is also true because the exploration for the regret minimization algorithm might be too slow.

## 2 Preliminaries

### 2.1 Notation

For a positive integer $K$, we use $[K]$ to denote $\{1, 2, \cdots, K\}$. For $r \in \mathbb{R}$, its absolute value is $|r|$, its ceiling integer is $\lceil r \rceil$, and its floor integer is $\lfloor r \rfloor$. Also, let $1_{\{.\}}$ be the indicator function.

### 2.2 Linear Bandits

We consider a linear bandit problem specified by a tuple $(\mathcal{X}, \theta)$, where $\mathcal{X}$ is the set of arms, and $\theta \in \mathbb{R}^d$ is an unknown but fixed parameter with $\|\theta\|_2 \leq 1$. Without loss of generality, we assume that $\mathcal{X} \subset \mathbb{R}^d$ and $\|x\| \leq 1$ for any $x \in \mathcal{X}$. At each round $t$, a learner chooses an action $x_t \in \mathcal{X}$ and observes the reward $r_t = \theta^\top x_t + \eta_t$, where $\eta_t$ is a symmetric and independent noise. We denote $x^* \in \operatorname{argmax}_{x \in \mathcal{X}} \theta^\top x$ as an optimal arm. Note that the linear setting includes the MAB problem as a special case. In the MAB setting, each arm $a \in \mathcal{X}$ is corresponding to a $|\mathcal{X}|$-dimensional standard unit vector (all entries are 0 except for the $a$-th entry).

In the online setting, i.e., an agent interacts with the bandit instance for at most $T \geq 1$ rounds, the performance of the agent is measured by the cumulative (pseudo) regret, which is defined by

$$\operatorname{Regret}(T) = \sum_{t=1}^{T} (\theta^\top x^* - \theta^\top x_t).$$

In order to obtain a small regret, the agent needs to figure out a vector close to $\theta$, even with the presence of the noise $\eta_t$. A related performance measure is the sample complexity of identifying a near-optimal arm. In this case, an agent is asked to output an arm $\widehat{x}$ such that

$$|\theta^\top x^* - \theta^\top \widehat{x}| \leq \epsilon$$

with probability at least $1 - \delta$. The sample complexity of the algorithm is measured by the smallest number of steps the agent interacts with the bandit instance. If Regret$(T)$ of an algorithm is sublinear, then we can use standard technique to covert it into an algorithm that outputs an $\epsilon$-accurate arm in time $T$ such that Regret$(T)/T = O(\epsilon)$. When $\epsilon = 0$, the problem reduces to the *best arm identification* problem. Note that best arm identification is usually not possible for a continuous action space $\mathcal{X}$.

### 2.3 Super Heavy-Tailed Noise

We now introduce the concept of characterizing the tail of the noise $\eta$.

**Definition 2.1** ($\alpha$-heavy-tail). We say a random variable, $\eta$, has $\alpha$-heavy-tail for some $\alpha > 0$, if $\alpha$ is the largest positive real number such that for all $y > 0$, we have $\Pr(|\eta| > y) \le \frac{1}{y^\alpha}$. A linear bandit instance, $(\mathcal{X}, \theta)$, is called with $\alpha$-*heavy-tail noise*, if for any arm $x \in \mathcal{X}$, its reward is a random variable $\theta^\top x + \eta$, for some *symmetrical* $\alpha$-heavy tail noise $\eta$.

**Remark 2.2.** For the random variable $\eta'$ such that $\Pr(|\eta'| > y) \le \frac{c}{y^\alpha}$ for some absolute constant $c > 0$, we have $\eta = \frac{\eta'}{c^{1/\alpha}}$ satisfies that $\Pr(|\eta| > y) \le \frac{1}{y^\alpha}$. Therefore, for ease of presentation, we assume $\Pr(|\eta| > y) \le \frac{1}{y^\alpha}$ here.

Note that the smaller the $\alpha$, the heavier the tail. Figure 1 shows several examples of the $\alpha$-heavy-tail noise. In particular, for a Cauchy random (Student's $t$-distribution with df $=$ 1.) variable $\eta$ with PDF, $f(x) = \frac{1}{\pi} \cdot \frac{1}{1+x^2}$, $\alpha = 1$; for a Student's $t(\text{df})$-distribution with PDF, $f(x) = \frac{1}{\sqrt{\text{df}} \cdot B(\frac{1}{2}, \frac{\text{df}}{2})} \cdot (1 + \frac{x^2}{\text{df}})^{-\frac{\text{df}+1}{2}}$, $\alpha = \text{df}$. The asymptotic behavior of $\alpha$-stable distribution is described by $f(x) \sim \frac{1}{|x|^{\alpha+1}}$, which shows that $\alpha$-stable distribution also has the $\alpha$-heavy tail. We also point out that normal distribution is not $\alpha$-heavy tail for any finite $\alpha$.

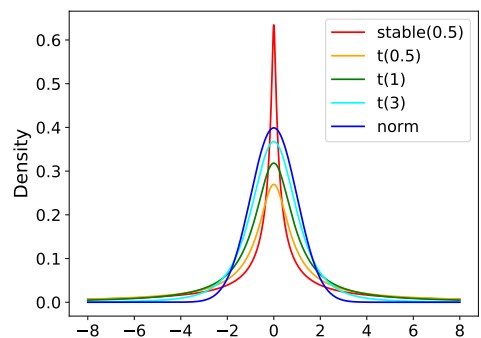

Figure 1: Probability density function (PDF) of normal distribution, 0.5-stable distribution, and Student's $t$-distribution with df = 0.5, 1, and 3.

Note that the symmetry assumption in the noise distribution is necessary. For $\alpha < 1$, the mean of $\eta$ is not necessarily existential. Hence, symmetry is needed for the problem to be well-defined. We note that for $\alpha > 1$, we can relax the assumption of symmetry by simply assuming $\mathbb{E}[\eta] = 0$. For the sake of presentation, we assume symmetry through out.

In comparison with existing literature, we require no assumptions on the bound or sub-Gaussianity of $\eta$, e.g., in (Abbasi-Yadkori et al., 2011). Meanwhile, we also impose no restrictions on the existence of the $(1 + \epsilon)$-th moment of the noise, which is required in Bubeck et al. (2013); Medina and Yang (2016); Shao et al. (2018); Xue et al. (2020). Indeed, the $\alpha'$-th moment of $\eta$ for any $\alpha' < \alpha$ does exist. However, when $\alpha \le 1$, previous algorithms fail to guarantee a sub-linear regret bound.

## 3 Robust Estimator for Super Heavy-tailed Noises

To introduce our estimator, we first briefly review previous robust estimators and provide some motivations of designing a new robust estimator. For rest of the section, we consider i.i.d. copies of random variable $X = x + \eta$, where $x$ is a fixed number and $\eta$ is a symmetric noise. Consider a sequence of $\tilde{n}$ i.i.d. copies of $X$: $X_1, X_2, \cdots, X_{\tilde{n}}$. There are two notable robust estimators for estimating $x$ in the heavy-tail setting.

The first robust estimator is the *truncated empirical mean* estimator, selects the random variables with magnitude smaller than certain threshold $c > 0$ and computes the empirical mean after selection:

$$\widehat{\mu}_1 = \frac{1}{\widetilde{n}} \sum_{i=1}^{\widetilde{n}} X_i 1_{\{|X_i| \leq c\}}. \tag{3.1}$$

The second one is called the *median of means*, which is defined as follows,

$$\widehat{\mu}_2 = \text{median}\left(\left\{\frac{1}{k}\sum_{i=1}^{k} X_{(j-1)k+i}\right\}_{j=1}^{k'}\right), \tag{3.2}$$

where $k$ is a parameter to be decided.

Unfortunately, as shown in Bubeck et al. (2013), these two robust estimators critically rely on the existence of mean of the random noise, and hence cannot be applied to super heavy-tailed noise with no mean. Indeed, the truncated random variable can significantly distort the center whereas the median of means estimator does not concentrate enough. We illustrate in more detail in Section 3.3 about their performance. Thus, for $\alpha$-heavy-tail random variables with $\alpha < 1$, new robust estimators are needed to effectively handle such noise.

## 3.1 Mean of Medians

The fundamental reason of the failure of the above estimators is due to the requirement of the existence of mean in the noise. To resolve this issue, we propose to use the empirical median, which is super robust against heavy tails as it characterizes the properties of the distribution, instead of moments. For instance, no matter how heavy the tail of the noise is, as long as it has certain probability of being close to 0, the median will have high probability being close to 0.

To leverage the above observation, we propose a novel statistical estimator: mean of medians (mom). Specifically, we split the $\widetilde{n}$ samples into $k'$ blocks and takes the mean of the median in each block. For $\widetilde{n}$ i.i.d. symmetric random variables $X_1, X_2, \cdots, X_{\widetilde{n}}$, we define the mean of medians estimator by

$$X_{\text{mom}} = \frac{1}{k'} \sum_{j=1}^{k'} \text{median}(X_{(j-1)k+1}, X_{(j-1)k+2}, \cdots, X_{jk}), \tag{3.3}$$

where $k = \lceil \widetilde{n}^\varepsilon \rceil$ and $k' = \lfloor \widetilde{n}/k \rfloor$. Here $\varepsilon \in (0,1)$ is a parameter depends $\epsilon$ and will be specified later.

## 3.2 Theoretical Guarantees of Mean of Medians Estimator

In this subsection, we provide the theoretical guarantees for the mean of medians estimator. First, we have the following lemma, which characterizes the median of heavy-tailed noises.

**Lemma 3.1.** Let $X_1, X_2, \cdots, X_m$ be $m$ i.i.d. symmetric random variables satisfying $\Pr(|X_i| > y) \leq \frac{1}{y^\alpha}$. Suppose $Y$ is the median of $X_1, X_2, \cdots, X_m$, we have

$$\Pr(|Y| \leq 4^{1/\alpha}) \geq 1 - 2e^{-m/8}.$$

*Proof.* First, we define the random variables $\widehat{X}_i = 1_{\{|X_i| > 4^{1/\alpha}\}}$ for $i \in [m]$. By the fact that $\Pr(|X_i| > y) \leq \frac{1}{y^\alpha}$, we have $p_i = \Pr(\widehat{X}_i = 1) \leq 1/4$. Together with Hoeffding's inequality, we obtain

$$\Pr\left(\sum_{i=1}^{m} \widehat{X}_i \geq m/2\right) \leq \Pr\left(\sum_{i=1}^{m} \widehat{X}_i - p_i \geq m/4\right) \leq e^{-m/8}.$$

Note that the median $Y$ satisfies $Y > 4^{1/\alpha}$ if and only if at least half of the estimates $X_i$ are above $4^{1/\alpha}$, which is equivalent to $\Pr(\sum_{i=1}^{m} \widehat{X}_i \geq m/2)$. Hence, we have $Y > 4^{1/\alpha}$ with probability at

most $e^{-m/8}$. Similarly, we can obtain that $Y < 4^{1/\alpha}$ with probability at most $e^{-m/8}$. Thus, it holds that

$$\Pr(|Y| > 4^{1/\alpha}) \leq 2e^{-m/8},$$

which concludes the proof of Lemma 3.1. $\square$

Lemma 3.1 shows that, when the number of samples $m$ is large, the median of $m$ i.i.d. super heavy tailed noise random variables is bounded by $4^{1/\alpha}$ with high probability. Equipped with this lemma, we formally describe the main results for the mean of medians estimator as follows.

**Theorem 3.2.** Let $\varepsilon, \alpha > 0$ be parameters. Let $X_1, X_2, \cdots, X_{\widetilde{n}}$ be $\widetilde{n}$ i.i.d. symmetric $\alpha$-heavy-tail random variables. When $\widetilde{n} \geq \max\{C, \left(16\log(2/\delta)\right)^{1/\varepsilon}\}$, for the mean of medians estimator defined in (3.3), we have

$$\mathbb{E}[X_{\mathrm{mom}}] = 0, \qquad \text{and} \quad |X_{\mathrm{mom}}| \leq \sqrt{\frac{2 \cdot 4^{2/\alpha}}{\widetilde{n}^{1-\varepsilon}} \cdot \log(4/\delta)}$$

with probability $1 - \delta$. Here $C$ is a constant depending on $\varepsilon$ such that $2C^{1-\varepsilon}e^{-C^{\varepsilon}/16} \leq 1$.

*Proof.* To facilitate our analysis, we denote the median of $(X_{(j-1)k+1}, X_{(j-1)k+2}, \cdots, X_{jk})$ by $Y_j$. Here $k = \lceil \widetilde{n}^{\varepsilon} \rceil$ and $k' = \lfloor \widetilde{n}/k \rfloor$. Under this notation, by Lemma 3.1, we obtain that

$$\Pr(|Y_j| \leq 4^{1/\alpha}) \geq 1 - 2e^{-k/8} \tag{3.4}$$

for any $j \in [k']$. Let $Z_j = Y_j \cdot \mathbf{1}_{\{|Y_j| \leq 4^{1/\alpha}\}}$, we have

$$\Pr(|X_{\mathrm{mom}}| > t) = \Pr\left(\left|\frac{1}{k'}\sum_{j=1}^{k'} Y_j\right| > t\right)$$

$$\leq \Pr\left(\left|\frac{1}{k'}\sum_{j=1}^{k'} Z_j\right| > t\right) + \sum_{j=1}^{k'} \Pr(|Y_j| > 4^{1/\alpha})$$

$$\leq 2\exp\left(-\frac{k't^2}{2 \cdot 4^{2/\alpha}}\right) + 2k'e^{-k/8},$$

where the last inequality follows from Hoeffding's inequality and Equation (3.4). When choosing

$$\widetilde{n} \geq \max\{C, \left(16\log(2/\delta)\right)^{1/\varepsilon}\},$$

where $C$ is a sufficient large constant depending on $\varepsilon$ such that $2C^{1-\varepsilon}e^{-C^{\varepsilon}/16} \leq 1$, we have that $2k'e^{-k/8} \leq \delta/2$. Hence, by setting $t = \sqrt{\frac{2 \cdot 4^{2/\alpha}}{\widetilde{n}^{1-\varepsilon}} \cdot \log(4/\delta)}$, we have

$$|X_{\mathrm{mom}}| \leq \sqrt{\frac{2 \cdot 4^{2/\alpha}}{\widetilde{n}^{1-\varepsilon}} \cdot \log(4/\delta)}$$

with probability at least $1 - \delta$. Together with the fact that the noise is symmetric, we conclude the proof of Theorem 3.2. $\square$

**Remark 3.3** (Sample Complexity). Solving the inequality $|X_{\mathrm{mom}}| \leq \zeta$ gives that $\widetilde{n} \geq \left(\frac{2 \cdot 4^{2/\alpha}}{\zeta^2} \cdot \log(4/\delta)\right)^{\frac{1}{1-\varepsilon}}$. Together with the constraint that $\widetilde{n} \geq \max\{C, (16\log(2/\delta))^{1/\varepsilon}\}$, we have $\widetilde{n} \geq \max\{C, (16\log(2/\delta))^{1/\varepsilon}, \left(2 \cdot \frac{4^{2/\alpha}}{\zeta^2} \cdot \log(4/\delta)\right)^{\frac{1}{1-\varepsilon}}\}$. If we choose $\varepsilon$ near to 0, $C$ and $(16\log(2/\delta))^{1/\varepsilon}$ are large. If we choose $\varepsilon$ near to 1, $\left(\frac{2 \cdot 4^{2/\alpha}}{\zeta^2} \cdot \log(4/\delta)\right)^{\frac{1}{1-\varepsilon}}$ is large. In other words, there is a trade-off in $\varepsilon$ to balance these three terms.

## 3.3 Comparison with Previous Robust Estimators

Coming back to the other robust estimators, truncated empirical mean and median of means have good guarantees when the mean of the random variables exist. Let us consider $\widetilde{n} > 0$ i.i.d. random variables $\{X_1, X_2, \cdots, X_{\widetilde{n}}\}$ and follow the notation in Bubeck et al. (2013). Specifically, suppose $X_i$ satisfies that

$$\mathbb{E}[X_i] = \mu, \quad \text{and} \quad \mathbb{E}[|X_i - \mu|^{1+\epsilon}] \le v, \quad \forall i \in [\widetilde{n}]$$

for some $\epsilon \in (0, 1)$. Bubeck et al. (2013) shows that, when we set $c$ and $k$ properly in (3.1) or (3.2), we have

$$|\mu - \widehat{\mu}| \lesssim v^{\frac{1}{1+\epsilon}} \left( \frac{C(\epsilon) \log(1/\delta)}{\widetilde{n}} \right)^{\frac{\epsilon}{1+\epsilon}}, \tag{3.5}$$

where $\widehat{\mu}$ is an estimator computed by (3.1) or (3.2). On the other hand, by Theorem 3.2, we have that the mean of medians estimator has an error rate of $\widetilde{O}(1/\widetilde{n}^{\frac{1-\varepsilon}{2}})$. In what follows, we compare the mean of medians estimator with previous robust estimators throughout two regimes of $\epsilon$.

- When $\epsilon \le 0$ [§], truncated empirical mean and median of means are not valid any more. In contrast, our estimator can still tackle such heavy tailed random variables.
- When $0 < \epsilon < 1$, by choosing $\varepsilon < \frac{1-\epsilon}{1+\epsilon}$ in (3.3), we know that mean of medians enjoys the convergence rate $\widetilde{O}(1/\widetilde{n}^{\frac{1-\varepsilon}{2}})$, which is better than the rate $\widetilde{O}(1/\widetilde{n}^{\frac{\epsilon}{1+\epsilon}})$ of truncated empirical mean and median of means.

For heavy tailed linear bandits, we mainly focus on the setting where $\epsilon < 1$ because Bubeck et al. (2013) proposes a nearly optimal algorithm with $O(\sqrt{T})$ regret when the noises have finite variance ($\epsilon \ge 1$). Therefore, we can conclude that our mean of medians estimator are preferable than previous robust estimator for heavy tailed linear bandits (especially for the super heavy-tailed linear bandits).

# 4 Generic Algorithmic Framework for Super Heavy-Tailed Linear Bandits

## 4.1 A Generic Bandit Algorithm

In this section, we propose a new algorithmic framework for the super heavy-tailed linear bandits defined in Definition 2.1. Specifically, by utilizing mean of medians (Algorithm 2) as a subroutine, we can transform any existing algorithm, e.g., (Dani et al., 2008; Rusmevichientong and Tsitsiklis, 2010; Abbasi-Yadkori et al., 2011; Soare et al., 2014; Jedra and Proutiere, 2020), into an efficient algorithm for super heavy-tailed linear bandits. The procedure is rather basic: the outer algorithm simply collects rewards of an arm and pass them into the mean of medians estimator, and use the output value as a new reward, which will have a light tail (by Theorem 3.2). The details are given in Algorithm 1.

---

**Algorithm 1** Synthetic Algorithm

---

1: **Input:** A bandit algorithm $\mathcal{A}$, $\delta > 0$, $\varepsilon > 0$ and an integer $\widetilde{n}$.
2: **for** $t = 1, 2, \cdots$ **do**
3:      Algorithm $\mathcal{A}$ chooses the arm $x_t$.
4:      Receive the reward $r_t \leftarrow$ Mean of Medians$(x_t, \widetilde{n}, \varepsilon)$. (Algorithm 2)
5: **end for**

---

---

**Algorithm 2** Mean of Medians

---

1: **Input:** An arm $y$ and an integer $\widetilde{n}$, and a parameter $\varepsilon \in (0, 1)$.
2: Set $k = \lceil \widetilde{n}^\varepsilon \rceil$ and $k' = \lfloor \widetilde{n}/k \rfloor$.
3: Pull the arm $y$ for $\widetilde{n}$ times and obtain the corresponding rewards $r_1, r_2, \cdots, r_{\widetilde{n}}$.
4: Let $Y_j$ be the median of $\{r_{(j-1)k+1}, r_{(j-1)k+2}, \cdots, r_{jk}\}$ for $j \in [k']$.
5: Set $r = \frac{1}{k'} \sum_{j=1}^{k'} Y_j$.
6: **return** $r$.

---

---

[§]Here $\epsilon < 0$ means that the mean of $X_i$ doesn't exist.

## 4.2 Theoretical Guarantees

In this subsection, we establish theoretical guarantees for our algorithmic framework (Algorithm 1).

**Theorem 4.1** (Regret Minimization). Suppose a linear bandit instance, $(\mathcal{X}, \theta)$, has $\alpha$-heavy-tail noise for some $\alpha > 0$. Fix $\varepsilon \in (0, 1)$. Let $\widetilde{n} = \lceil \max\{C, \left(16 \log(2T/\delta)\right)^{1/\varepsilon}, (2 \cdot 4^{2/\alpha} \log(4/\delta))^{\frac{1}{1-\varepsilon}}\} \rceil$, $C$ is a constant depending on $\varepsilon$ such that $2C^{1-\varepsilon} e^{-C^{\varepsilon}/16} \leq 1$ in Algorithm 1. Let $\mathcal{A}$ be a linear bandit algorithm, which achieves regret bound $R(d, T, \delta)$ under 1-sub-Gaussian noises with probability at least $1 - \delta$. Then, Algorithm 1 with input $(\mathcal{A}, \delta, \varepsilon, \widetilde{n})$ enjoys a regret bound

$$\widetilde{n} \cdot R(d, T/\widetilde{n}, \delta)$$

with probability at least $1 - 2\delta$.

*Proof.* Fix $T > 0$. We divide total $T$ steps into $\widetilde{T} = \lfloor T/\widetilde{n} \rfloor$ blocks, each of which consists of $\widetilde{n}$ steps. With each block, we pull an arm for $\widetilde{n}$ times. For $t$-th block, we assume that $r_t = \theta^\top x_t + \eta_t$. By Theorem 3.2, for any $t \in [T]$, we have

$$\mathbb{E}[\eta_t] = 0, \quad \text{and} \quad |\eta_t| \leq 1$$

with probability at least $1 - \delta/T$. Thus, we reduce the super heavy tailed linear bandits into linear bandits with 1-sub-Gaussian noises. Together with our assumption that $\mathcal{A}$ achieves regret $R(d, T, \delta)$ for the 1-sub-Gaussian linear bandits with probability at least $1 - \delta$, we have

$$\text{Regret}(T) \leq \widetilde{n} \cdot R(d, T/\widetilde{n}, \delta),$$

with probability at least $1 - 2\delta$, which concludes the proof of Theorem 4.1. $\qquad \square$

**Remark 4.2.** The parameter $\alpha$ does not need to be known exactly. Any lower bound of the true $\alpha$ suffices to ensure the same guarantee. It can also be treated as a hyper parameter in the algorithm.

**Remark 4.3.** Shao et al. (2018); Xue et al. (2020) establish an expected regret lower bound $\Omega(T^{\frac{1}{1+\epsilon}})$ for the linear bandits with heavy-tailed payoffs, where the payoffs admit finite $1 + \epsilon$ moments for some $\epsilon \in (0, 1]$. This is not inconsistent with our conclusion since we consider the pseudo regret instead of the expected regret. Moreover, it is reasonable to consider the pseudo regret since we cannot define the expected regret when the mean of the noise does not exist.

**Remark 4.4.** Note that for any fixed $\alpha > 0$, $\widetilde{n}$ is only logarithmically depending on $T$. Hence the overall regret bound is only a factor of $\text{poly} \log(T)$ worse compared to the light tail counterpart. If algorithm $\mathcal{A}$ obtains a near-optimal regret for sub-Gaussian noise (e.g., algorithms in Abbasi-Yadkori et al. (2011)), then our regret bound is near optimal for $\alpha$-heavy-tail noise as well. To instantiate Algorithm 1, we apply the near-optimal algorithm in Abbasi-Yadkori et al. (2011), and immediately obtain the following near-optimal regret bound.

**Corollary 4.5.** Suppose a linear bandit instance, $(\mathcal{X}, \theta)$, has $\alpha$-heavy-tail noise for some $\alpha > 0$. We use the OFUL algorithm in Abbasi-Yadkori et al. (2011) as the input in Algorithm 1 and set $\varepsilon = 1/2$. Let $\widetilde{n} = \lceil \max\{C, \left(16 \log(2T/\delta)\right)^2, (2 \cdot 4^{2/\alpha} \log(4/\delta))^2\} \rceil$, where $C$ is a constant such that $2\sqrt{C} e^{-\sqrt{C}/16} \leq 1$ in Algorithm 1. Then, by using Algorithm 1 achieves a regret bound $\widetilde{O}(d\sqrt{\widetilde{n}T} \log(T/\delta))$ with probability at least $1 - \delta$.

## 5 Experiment

In this section, we conduct numerical experiments to demonstrate the effectiveness of our algorithmic framework for super heavy-tailed linear bandit problems. Experiments are run in a Windows 10 laptop with Intel(R) Core(TM) i7-8750H CPU and 16GB memory.

We adopt state-of-the-art algorithms, i.e., SupBMM and SupBTC, proposed by Xue et al. (2020) as the input algorithm $\mathcal{A}$ in Algorithm 1, yielding synthetic algorithms SupBMM_mom and SupBTC_mom respectively. Here "mom" means using our mean of medians estimator to process noise as in Algorithm 1. Apart from SupBMM and SupBTC of Xue et al. (2020), we also make comparisons with MoM and CRT of Medina and Yang (2016), MENU and TOFU of Shao et al. (2018). All the traditional algorithms require moments condition, i.e., for some $\epsilon \in (0, 1]$, $(1 + \epsilon)$-th central moment of the heavy-tailed noise is bounded under some $v > 0$.

Figure 2: Comparison of our algorithms versus MoM, CRT, MENU, TOFU, SupBMM and SupBTC in heavy-tailed linear bandit problems for $1 \times 10^4$ rounds.

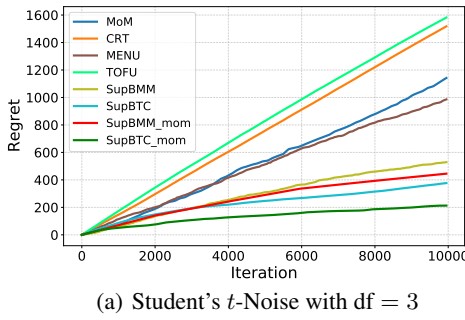
(a) Student's $t$-Noise with df $= 3$

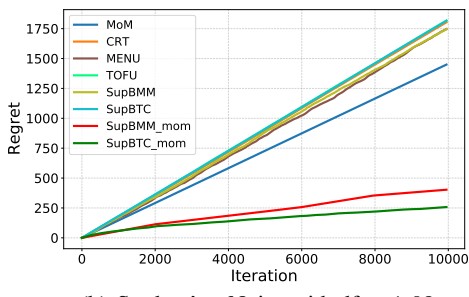
(b) Student's $t$-Noise with df $= 1.02$

Figure 3: Comparison of our algorithms versus MoM, CRT, MENU, TOFU, SupBMM and SupBTC in super heavy-tailed linear bandit problems for $1 \times 10^4$ rounds.

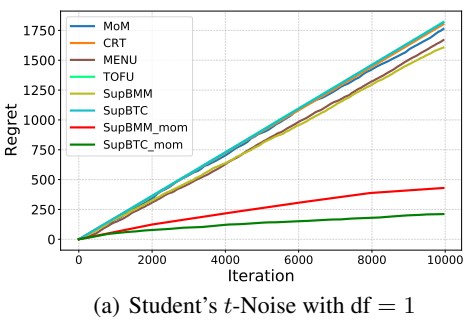
(a) Student's $t$-Noise with df $= 1$

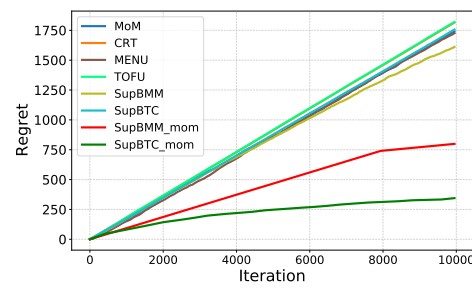
(b) Student's $t$-Noise with df $= 0.5$

For comparison, we show cumulative regret with respect to number of rounds of bandits played over a fixed finite-arm decision set $\mathcal{X}$. We generate 10 independent paths for each algorithms and show the average cumulative regret. We use the following experimental setup corresponding to that in Medina and Yang (2016); Xue et al. (2020). Let the feature dimension $d = 10$, the number of arms $|\mathcal{X}| = K = 20$. For the chosen arm $x_t \in \mathcal{X}$, reward is $\theta^{*\top} x_t + \eta_t$, where $\theta^* = \mathbf{1}_d / \sqrt{d} \in \mathbb{R}^d$ so that $\|\theta^*\|_2 = 1$ and $\eta_t$ is sampled from a Student's $t$-distribution with degree of freedom df as a parameter to be specified. Every contextual information of time $t$, i.e., $x_{t,a}, a \in [K]$, is sampled from uniform distribution of $[0, 1]$ respectively for each dimension, with normalization made to ensure $\|x_{t,a}\|_2 = 1$.

This section is divided into 2 parts: First, we choose an environment with heavy-tailed noise, whose $(1 + \epsilon)$-th central moment is finite for some $\epsilon \in (0, 1]$. Student's $t$-noises with df $\in \{3, 1.02\}$ are chosen, whose moment parameter $\epsilon \in \{1, 0.01\}$ and bound parameter $v \in \{3, 65.19\}$ respectively. Then we consider linear bandits with super heavy-tailed noise. Student's $t$-noises with df $\in \{1, 0.5\}$ are chosen. In this setting, $(1 + \epsilon)$-th central moment no longer exists for any $\epsilon \geq 0$. In theory, none of the algorithms mentioned above could work properly. In order to make other algorithms work, we input $\epsilon = 0.01$ and treat $v$ as a hyper parameter that needs to be tuned for relatively good performance. We remark that Algorithm 1 is relatively not sensitive to the choice of $v$ (See appendix).

Our algorithm's parameter $\varepsilon$ is set to $0.5$ since results vary little with $\varepsilon$ empirically (See appendix for more information). And $\widetilde{n}, k, k'$ is set according to Theorem 4.1 and Algorithm 2. For the noise processed by our mean of medians estimator (Algorithm 2), we input $\epsilon = 1$ and tune $v$ to ensure the performance of Algorithm 1. Specifically, no matter how heavy-tailed original noise is, our mean of medians estimator can reduce super heavy-tailed noise to bounded noise with high probability, as is shown in Theorem 3.2. So it is reasonable to assume the processed noise has a finite second moment, which is bounded by an unknown $v$ to be tuned, i.e. $\mathbb{E}[|\eta_{\mathrm{mom}}|^2] \leq v$.

We show experimental results of heavy-tailed linear bandit problems in Figure 2 for $1 \times 10^4$ rounds. Figure 2(a) compares our algorithms with the aforementioned six algorithms under Student's $t$-noise with df $= 3$, which corresponds to the results in Xue et al. (2020). Figure 2(b) presents regret versus iteration with df $= 1.02$. In Figure 2(b), our algorithms outperform MoM, CRT, MENU, TOFU, SupBMM and SupBTC with heavy-tailed noise as expected. Specially, in Figure 2(b), df $= 1.02$, so 1.01-th central moment exists, which is bounded by a rather big number $v = 65.19$. In this setting, all of other algorithms perform poorly, whereas our algorithms work perfectly well, which verifies effectiveness of our algorithms as in Section 3.3 and Theorem 4.1.

Experimental results of super heavy-tailed linear bandit problems are demonstrated in Figure 3 for $1 \times 10^4$ rounds. Figure 3(a) and (b) consider df $= 1, 0.5$ respectively. As is shown in Figure 3, our algorithms perform significantly better than other algorithms. And SupBTC_mom performs better than SupBMM_mom.

In a word, our algorithms outperform the state-of-the-art algorithms even when $\epsilon > 0$, and have comparably good performance when $\epsilon \leq 0$, which is consistent with the theoretical results in Theorem 4.1.

## 6    Conclusion

In this work, we have proposed a generic algorithmic framework for super heavy-tailed linear bandits. Such an algorithmic framework incorporates a classical linear bandit algorithm to tackle existing challenges such as the trade-off between exploration and exploitation, and more importantly, adopts the mean of medians estimator to handle the challenge of super heavy-tailed noises. We show that our algorithmic framework is provably efficient for regret minimization. Meanwhile, we conducted numerical experiments to validate the effectiveness of our framework in practice. To the best of our knowledge, we make the first attempt to study the super heavy-tailed linear bandits and propose the first provably efficient method that successfully handles super heavy-tailed noises.

## 7    Acknowledgments

The authors would like to thank anonymous reviewers for their valuable advice. Part of the work was done while Han Zhong and Jiayi Huang were students in University of Science and Technology of China. This work was supported by National Key R&D Program of China (2018YFB1402600), Key-Area Research and Development Program of Guangdong Province (No. 2019B121204008), BJNSF (L172037) and Beijing Academy of Artificial Intelligence. Project 2020BD006 supported by PKU-Baidu Fund.

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
