# OpenReview forum: "Breaking the Moments Condition Barrier: No-Regret Algorithm for Bandits with Super Heavy-Tailed Payoffs"
_NeurIPS.cc/2021/Conference — NeurIPS 2021 Poster_

### Official Review · Reviewer_uiUt · 2021-07-15

**Rating:** 7
**Confidence:** 4

**Summary:**

The paper makes the first attempt to handle symmetric super heavy-tailed noise in a linear
bandit setting. A new estimator called mean-of-medians is proposed to estimate parameters of a
linear model under heavy-tailed noise. A generic algorithmic framework for solving linear bandit
problems is proposed where the mean-of-medians estimator can be plugged in and existing
high-probability regret bounds for sub-gaussian arms can be used to get high-probability
regret bounds for the proposed setting. Numerical experiments are also provided.

**Limitations And Societal Impact:**

1. Not reporting error bars or a measure of variability in a heavy-tails paper doesn't seem right.
I don't understand why the authors have chosen to say that 'error bars are not important here.'
I understand that the regret bounds hold with high probability and the expected regret doesn't exist.
Therefore, it is even more important to let the reader know that what you claim works for many
sample paths and not just a single sample path, like it is done here. I couldn't find any details
about this even in the appendix.

2. I am not comfortable with the idea of tuning transformed noise variance ($\nu$).
This is reasonable to do in a sub-Gaussian setting where there is a lot of data and deviations
are not very large. With heavy-tails/rare-events, where by definition, deviations are large
and useful data is sparse, this doesn't seem right. I would like to see how sensitive
your mom-algorithms are to the choice of $\nu.$

3. Please address my concerns about applicability of this work (see 'Significance' above).

-----

My concerns have been satisfactorily addressed by the authors.

**Main Review:**

*Originality*: To my knowledge, this is the first time that a paper has considered super heavy-tailed
noise in the bandit setting. The techniques in the paper are simple but work out.
I liked how the analysis for mean-of-medians estimator works out with the symmetric noise
assumption. I think a few more papers should be cited: [1], [2], and [3].
I believe these papers were among the first few to consider heavy tails in the regret setting and the
fixed-confidence setting.

*Quality*: The theoretical results are easy to understand but I have concerns about how the
experiments are conducted. Details are in the limitations section.

*Clarity*: The theoretical arguments are clear. I have made suggestions in next section on how
experiments can be made more informative.

*Significance*: I think dealing with heavy-tails is difficult and all bets are off if considering super
heavy-tails. There is a requirement of restrictive assumptions like symmetric noise to do any
analysis. Practically, the skew plays a big role in fields like finance, see [4]. I am not sure
in what applications the assumption of symmetric super-heavy tails is valid and doesn't
cause blow-ups.

I would consider the work more significant if primarily my concerns about the experiments are addressed.

[1] Vakili, Sattar, Keqin Liu, and Qing Zhao. "Deterministic sequencing of exploration and exploitation for multi-armed bandit problems." IEEE Journal of Selected Topics in Signal Processing 7.5 (2013): 759-767.
[2] Carpentier, Alexandra, and Michal Valko. "Extreme bandits." Neural Information Processing Systems. 2014.
[3] Glynn, Peter, and Sandeep Juneja. "Selecting the best system and multi-armed bandits." arXiv preprint arXiv:1507.04564 (2015).
[4] Mandelbrot, Benoit B., and Richard L. Hudson. The (mis) behaviour of markets: a fractal view of risk, ruin and reward. Profile books, 2010.

**Time Spent Reviewing:**

4 hours

---

> ### Author Response · Authors · 2021-08-10
> **Response to Reviewer uiUt:**
>
> Thank you for being positive about the paper. Thanks also for pointing out some related papers. We will cite these papers in the next revision.
>
> Thanks for raising the points about the skewness in practice. The current analysis relies on the symmetric assumption merely due to the definition of regret. Indeed we can consider non-symmetric noise. However, we still need a notion of “center” in the distribution to have a well-defined regret. For instance, if the moment's condition holds, the center is the mean. If the distribution is symmetric, then the center is the median. Indeed, symmetry is not a necessary condition. It will be our next step to figure out more conditions that the estimator is still working.
>
> Regarding symmetric heavy-tailed noise, there are many examples in practice, e.g., [Rachev, 2003] and [Godsill, S., & Kuruoglu, E. E. (1999)]. In [Rachev, 2003], the random variable $X$ with $Pr(|X| > x) \sim x^{-\alpha} L(x)$ is usually considered in a finance application, where $L$ is a slowly varying function. And the super heavy-tailed random variable defined in our paper can be regarded as a special case of this type of random variable. For [Godsill, S., & Kuruoglu, E. E. (1999)], $\alpha$-stable distributed noise is considered in time series inference. We will emphasize this point more in the next version.
>
> Single sample path v.s. multiple sample paths: Thanks for raising this point. In our experiment, we drew a single path for each algorithm. The sample path contains 10000 iterations and is very smooth, which indicates the high-probability behavior as proved by our theorems. We thus do not include multiple paths in the paper.
>
> In fact, we have generated 10 sample paths and compared them. Each one of them shows very similar behavior. We will include all these figures in the next revision. Since a sample path contains a lot of points, the error bar (computed using the single path) becomes too small to show (hence not important). You are indeed right that if we draw all the paths on the same figure, it is important to include the variation range. We will add this in the next revision.
>
> Noise variance $\nu$: Did you mean the parameter $v$ in the experiment? Note that the noise transformed by the mean of the median algorithm is bounded with high probability (Theorem 3.2), which allows us to calculate $v$. The idea of tuning the parameter $v$ is only for better performance. Moreover, we remark that the mom-algorithms are relatively not sensitive to $v$ since the median estimator is very stable even for very rare events. We will clarify this in the revision.
>
>
> [Svetlozar Todorov Rachev. Handbook of Heavy Tailed Distributions in Finance: Handbooks in 372 Finance, Book 1. Elsevier, 2003.]
>
> [Godsill, S., & Kuruoglu, E. E. (1999). Bayesian inference for time series with heavy-tailed symmetric α-stable noise processes. Proc. Applications of heavy tailed distributions in economics, engineering and statistics.]

---

> > ### Comment · Reviewer_uiUt · 2021-08-25
> > **Request for plots**
> >
> > Can you please share the plots here in a manner which respects double-blind reviewing?
> > I just want to see your two MOM algorithms and only for t-noise with df=0.5.
> > Please select three values of parameter $v$ over a suitably large range.
> >
> > I think the experiments are inexpensive to run. Can you increase the number of sample paths to 100 or 500?
> > Please report the average and median regret across sample paths.
> >
> > I am requesting for 6 sets of experiments and 12 curves. I hope it is not too much.

---

> > > ### Author Response · Authors · 2021-08-27
> > > **Response to Reviewer uiUt with plots**
> > >
> > > Thanks for your comments and suggestions about the experiment. We share the plots anonymously in the link below. https://ufile.io/bvah56x5
> > >
> > > We hope this will be helpful to you.

---

> > > > ### Comment · Reviewer_uiUt · 2021-08-27
> > > > **I have increased my score to 7**
> > > >
> > > > Thank you for sharing the plots. I have increased my score to 7.
> > > >
> > > > I think there was some value in making you share the plots. SupBTC_mom seems to be doing as good as the plots in the paper but SupBMM_mom does visibly worse. I hope I am not mistaken. Please provide some explanation on these findings at least in the appendix of the revised version. I would also suggest that you report either the average or the median regret in the main version instead of regret of just one run.

---

> > > > > ### Author Response · Authors · 2021-08-27
> > > > > **Thanks for your re-evaluation**
> > > > >
> > > > > Thanks for your re-evaluation and useful suggestion. We will double-check our experiment and incorporate your suggestion in the next revision of our paper.

---

### Official Review · Reviewer_fcbK · 2021-07-16

**Rating:** 7
**Confidence:** 4

**Summary:**

The paper addresses the problem where the noise has super-heavy tail, in the framework of online learning. In the previous works, it is assumed that the 1+\epsilon moment of the noise exists, where $\epsilon >0$. However, in this paper, a polynomially decaying tail bound is only assumed, which coves the setting, when the first moment does not exist. Hence, the previous methods like median of means cannot be used. Instead, here the authors develop mean of medians algorithm. With this, the paper first converts the super-heavy tailed noise to a bounded one, and then use this noise to play any online (ex. linear bandit) algorithm. Even when 1+\epsilon moment exists, a comparison study is also provided. Furthermore, the theory is corroborated with experiments.

**Ethical Concerns:**

No ethical concerns.

**Limitations And Societal Impact:**

Limitations mentioned briefly. No societal impacts mentioned. See above for limitations.

**Main Review:**

The paper is very well-written. The theoretical novelty is clear. Comparison with previous works are also provided. The approach is innovative.

It would be great if the authors could properly motivate practical settings (there are a few mentions in the paper), where this kind of noise is relevant. Can you give an example where the noise mean doesnot exist (ex. cauchy noise) in an application? The second moment onwards have some potential applications mentioned in the previous papers.

Also, is it possible to get back the results with (conditionally) sub-Gaussian noise with choices of \alpha?

Also, how a lower bound on \alpha is found in practice would be of interest. Please comment on this.

Also, remark on the \alpha dependence of the regret expression.

The experiments are  adequate.

**Time Spent Reviewing:**

3

---

> ### Author Response · Authors · 2021-08-10
> **Response to Reviewer fcbK:**
>
> Thanks for your appreciation of the paper.
>
> 1. Application: In finance, the random variable $X$ with $Pr(|X| > x) \sim x^{-\alpha} L(x)$ is usually considered, where $L$ is a slowly varying function. The super heavy-tailed random noise defined in our paper can be regarded as a special case of this type of random noise. See [Rachev, 2003] for details. $\alpha$-stable random noises are also widely considered in statistics, e.g., [Godsill, S., & Kuruoglu, E. E. (1999)].
>
> 2. If the noise has finite $m$-th moment (e.g., sub-Gaussian), then by Markov inequality we have this noise is $m$ super heavy-tailed. Hence our result can be applied and gives similar regret bounds as in previous results (up to constant factor).
>
> 3. We can find $\alpha$ by the doubling trick. For example, starting with $\alpha=1$, if each time we have an estimator with too large empirical variation, we decrease $\alpha$ by $\alpha\gets\alpha/2$. We will clarify this more in the next version.
>
> 4. Dependence of $\alpha$: Note that our regret is $\tilde{n} \cdot R(d, T/\tilde{n}, \delta)$, which depends on $\tilde{n}$, whose dependence of $\alpha$ is formally defined in Theorem 4.1. Informally,  for big $\alpha>1$, $\tilde{n}$ is about a constant, and the regret bound tends to have no dependence on $\alpha$. For small $\alpha \to 0$, the regret bound is approaching linear in $T$.
>
> [Svetlozar Todorov Rachev. Handbook of Heavy Tailed Distributions in Finance: Handbooks in 372 Finance, Book 1. Elsevier, 2003.]
>
> [Godsill, S., & Kuruoglu, E. E. (1999). Bayesian inference for time series with heavy-tailed symmetric α-stable noise processes. Proc. Applications of heavy tailed distributions in economics, engineering and statistics.]

---

> > ### Comment · Reviewer_fcbK · 2021-08-16
> > **Response to comments**
> >
> > I thank the authors for their response. I have also gone through the concerns raised by other reviewers and the response to them. I have the following comments:
> >
> > 1) I had concerns on the practical examples and a few technical points, and based on the response of the reviewers, I am satisfied (thanks for the reference).
> >
> > 2) Reg. concerns raised by Reviewer 4arc, it is not clear that the median alone suffices with symmetric noise. The authors have responded to it adequately. A discussion on this is I believe worthy in the revised version.

---

> > > ### Author Response · Authors · 2021-08-17
> > > **Thanks for the response**
> > >
> > > Thank you for reading our response. We will incorporate your suggestion in the next revision of our paper.

---

### Official Review · Reviewer_hCX4 · 2021-07-16

**Rating:** 8
**Confidence:** 4

**Summary:**

The paper gives a novel robust estimator -- mean of medians -- for heavy-tailed noise satisfying $Pr( |Y| > |y|) \leq |y|^{-\alpha}$ for any fixed $\alpha > 0$. In particular, given access to i.i.d. samples $X_1,\ldots,X_n$, the unbiased estimator returns $X_{\sf mom}$ satisfying (i) $E [X_{\sf mom}] = 0$ and (ii) $|X_{\sf mom}| \leq \sqrt{4^{2/\alpha}/n^{1 - \epsilon} \log(4/\delta)}$ with probability at least $1 - \delta$. The paper then uses this as a black box to give a $\tilde{O}(d\sqrt{T})$-regret algorithm for linear bandits in $d$-dimension.



**Limitations And Societal Impact:**

Yes

**Main Review:**

Heavy tailed noise reward structures are a relatively underexplored area in MAB, there have only been a handful of previous works (e.g., see [1,2]) which address this for specific classes. The main contribution of this paper is that it proposes a novel robust estimator for heavy tailed noise : mean of medians, which is then used as a black box in off-the-shelf linear bandit algorithms. This paper introduces a novel robust estimator -- the mean of medians estimator -- for a general family of heavy tailed noise distributions which satisfy $Pr[|Y| \geq |y|] \leq |y|^{-\alpha}$ for some $\alpha > 0$. In particular, it shows that the estimator error goes to 0 at the rate of $\sqrt{4^{2/\alpha}/n^{1 - \epsilon}}$, whenever $n \geq n(\epsilon)$, for any $\epsilon > 0$. This combined with [3] immediately gives a $\tilde{O}(d\sqrt{T})$ regret algorithm for linear bandits with heavy tailed noise of the above form. The results are also supported using empirical valuations.

The authors of the paper identify that the key shortcoming of previous estimators in this setting require the existence the of the mean. In this paper, this obstacle is overcome by first computing the medians $m_1...,m_{n/k}$ of equipartitioned $k$-buckets of the observations ${(X_{(j-1)i + 1},\ldots,X_{ij})}_{i \in [n/k]}$,  and then outputting the mean of truncated medians. The key observation here is that as long as the distribution assigns non-trivial mass in a neighborhood around $0$, the median will also tend to $0$ with non-trivial probability. Combining this with a standard application of Hoeffding's on the truncated medians establishes the concentration for the estimator.

In summary, this is a nice clean result with useful applications.

Strengths
--------------------------------
1. Studies a important but relatively underexplored problem in MAB.
2. Proposed solution is conceptually clean and elegant.

Weaknesses
--------------------------------
1. Establishing the tail bound is not very involved, given the estimator, so I am not sure if this would be difficult to establish for experts in this area.

Question for the Author(s):
--------------------------------
Is there a family of distributions for which estimator is sample complexity optimal?

[1] Bubeck, S., Cesa-Bianchi, N. and Lugosi, G., 2013. Bandits with heavy tail. IEEE Transactions on Information Theory, 59(11), pp.7711-7717.

[2] Medina, A.M. and Yang, S., 2016, June. No-regret algorithms for heavy-tailed linear bandits. In International Conference on Machine Learning (pp. 1642-1650). PMLR.

[3] Abbasi-Yadkori, Y., Pál, D. and Szepesvári, C., 2011. Improved algorithms for linear stochastic bandits. Advances in neural information processing systems, 24, pp.2312-2320.


**Time Spent Reviewing:**

4

---

> ### Author Response · Authors · 2021-08-10
> **Response to Reviewer hCX4:**
>
> Thank you for your time and appreciation of the paper. We agree that “establishing the tail bound” is not involved. However, as you also pointed out, the contribution of the paper is about the novel estimator but not its technical difficulty.
>
> Our estimator is nearly regret-optimal in terms of T. However, the dependence of $\alpha$ is $4^{1/\alpha}$. Hence it is near-optimal for $\alpha$ bounded away from 0. For very small $\alpha$, we leave it as an open question for whether a better estimator exists. Thanks for raising this question.

---

### Official Review · Reviewer_4arc · 2021-07-24

**Rating:** 6
**Confidence:** 4

**Summary:**

This paper addresses Multi-Armed Bandit (MAB) problems when rewards are "super heavy-tailed", meaning when the mean (first order moment) might not exist. Recall that previous works [1] usually require the existence of a moment of order $1 + \epsilon$, with $\epsilon > 0$. To address this problem, the authors introduce the Mean-of-Medians estimator, which is shown to concentrate well even in the super heavy-tailed framework (Theorem 3.2). This estimator can then be plugged in existing algorithm (Theorem 4.1). Numerical experiments are provided, showing the advantage of the proposed method over previous works (Section 5).

[1] "Bandits with heavy tail" Bubeck et al. 2013

**Limitations And Societal Impact:**

Yes

**Main Review:**

The topic of the paper is of great interest, but I am not convinced by the answered proposed in the present article.

In particular, I can't see any motivation for the Mean-of-Medians estimator introduced. If for the Median-of-Means the idea is rather clear (the median operator allows to robustly aggregate the means, that can poorly concentrate without subGaussianity), the story is not clear for the Mean-of-Median. So my first question is: why not simply considering the median?

My feeling is that, by considering symmetric noise (an assumption that can be argued), the median (or the Mean-of-Medians) are indeed good estimators of the center of the distribution, that work even if the expectation is not defined. Without the moment condition, other estimators fail, but can't the success of Mean-of-Medians be explained entirely by the symmetry assumption?

The proof of Theorem 3.2 contains several points that should be clarified:
- last display equation before l. 208: shouldn't it be $4^{\frac{2}{\alpha} + 1}$ at the denominator, as Hoeffding is used with $a = 4^{1/\alpha}$ and $b = -4^{1/\alpha}$?
- l. 209: the choice $C=0$ always works. Beware that the function in $C$ presented is not monotonic, so taking $n  > C$, for a $C$ satisfying the inequality does not mean that $n$ also satisfies the inequality.
- the computations are made with real values for $k$ and $k'$, a more detailed proof considering the flooring should be added
- is it expected that the best rate is obtained for $\varepsilon = 0$, which means $k = 1$, and the Mean-of-Medians becomes the empirical mean? Then, what is the advantage of Mean-of-Medians?
- I feel that the dependence of $C$ with respect to $\varepsilon$ should be better discussed, in particular since this quantity is supposed to multiply the regret guarantees (Theorem 4.1). For $\varepsilon = 0.2$, I got $C = 10^{13}$.

More generally, the paper needs a important polishing:
- l. 33: $d$ is not defined
- l. 36: indentify*ing a* best arm
- l. 42: second *order* moment
- l. 121: co*n*vert it *into*
- l. 136-141 are not clear at all
- the presentation of Median-of-Means (eq. 3.2) is not very clear for someone who is not already familiar with it. And $k$ does not depend on the property of the random variable, only on the confidence level wanted (l. 167)
- l. 176: to use *the* empirical median
- l. 180: *we* propose
- l. 180: MoM is an acronym which is widely spread for Median-of-Means, it is very confusing to define it for Mean-of-Medians
- l. 182: why using $\tilde{n}$ and not just $n$?
- l. 183: the use of $\varepsilon$ and $\epsilon$ is confusing, all the more that they are related
- l. 183: (and several times, e.g. l. 204, 209) either *which depends on* or "depending on"
- l. 201: $\varepsilon$ and not $\epsilon$ I assume
- using $k$ and $k'$ for the number of blocks / size of the blocks is also confusing

Overall, despite the interest of the problem addressed in the present article, I think the present contribution is not ready for publication. Beyond a global polishing, it needs to motivate more the use of the Mean-of-Medians, especially against the standard mean and the standard median. Given the symmetry assumption, the latter seems to provide a natural alternative. The roles of $\varepsilon$ and $C$ should be better detailed.

**Time Spent Reviewing:**

5

---

> ### Author Response · Authors · 2021-08-10
> **Response to Reviewer 4arc:**
>
> We thank the reviewer for raising the concerns. However, we disagree with the proposal that a simple median would solve the problem. This is NOT true. Consider the case where the noise is either -1, or 1 and the distribution is symmetric. Then the median (for an odd number of samples) is either 1 or -1. Such an estimator can never be any closer to the ground truth, 0.
>
> In our settings, the mean of medians is inevitable. With our design, the median estimator converts the heavy-tailed random noise to a light-tailed one. The means estimator provides the concentration. Hence, both pieces are important in the design.
>
> The success of our new estimator is definitely not due to the symmetry property. Nevertheless, we stress that the symmetry assumption is natural. The previous median-of-means estimator works under the assumption that the expectation is 0, which is another kind of symmetry. Heavy tailed symmetric noise is also often in practice, e.g., stable distribution [Godsill, S., & Kuruoglu, E. E. (1999)]. Even with this condition, no previous solutions can ever handle the heavy-tailedness. Moreover, the symmetry assumption is to ensure that the mean of the final estimator $X_{mom}$ is zero (see Theorem 3.2), which might not be necessary. There might be other conditions for the estimator to work. We leave this for future work.
>
> We thank the reviewer for pointing out the typos. Here we clarify some of the confusion.
> 1. Last display equation before l.208: Yes, you are right. Thanks for pointing this out.
> 2. Why we cannot choose $\varepsilon = 0$: In our algorithm, $\varepsilon$ should be some positive constant to ensure that $k’ e^{k/8}$ (Line 208) is small for some sufficiently large $\tilde{n}$. When $\varepsilon=0$, the mom estimator becomes simply the median estimator, which does not work as we discussed above.
> 3. The choice of $C$ in l.209: $C$ is chosen to be a sufficiently large constant that $k’ e^{k/8}\le \delta/2$. Hence it cannot be 0. Moreover, the function is monotonic for sufficiently large $C$, which guarantees that $\tilde{n}$ also satisfies the inequality. We will clarify this more in the revision.
> 4. We have considered the flooring. See l.206 for the definitions of $k$ and $k’$.
> 5. Thanks for pointing it out for the presentation of the median of means estimator (Eq 3.2). In the next version, we will introduce the mean of medians estimator in more detail.
> 6. We use the acronym ‘‘mom’’ instead of ‘‘MoM’’ throughout this paper (including Line 180).
> 7. The relationship of $\alpha$, $\varepsilon$, and $C$ are specified in Theorem 3.2 and 4.1. We can indeed optimize $\varepsilon$ such that $\tilde{n}$ is at its minimum. However, the exact solution for $\varepsilon$ is hard to obtain. For an approximate solution, we can always choose, e.g., $\varepsilon=1/2$. Then $C$ is simply a large constant integer. In this case, $\tilde{n}$ is of the order $4^{4/\alpha}\cdot \text{poly}\log(T/\delta)$.
>
> Thanks for your other suggestions. We will carefully polish our paper accordingly in the revision.
>
> [Godsill, S., & Kuruoglu, E. E. (1999). Bayesian inference for time series with heavy-tailed symmetric α-stable noise processes. Proc. Applications of heavy tailed distributions in economics, engineering and statistics.]

---

> > ### Comment · Reviewer_4arc · 2021-08-30
> > **.**
> >
> > I thank the authors for their response.
> >
> > - I agree that in the proposed example the Mean-of-Medians might behave better than the simple median. It should definitely be incorporated in the core text as a part of some motivation for the proposed estimator (that I feel is missing in the current version). However, have you benchmarked the median solution, e.g., for experiments of Figure 2.a ? I was not able to run the provided code due to a missing file (./data/mom_10000_student_3_noise.txt)
> >
> > - if the symmetry is not important, I would suggest to simply remove this assumption
> >
> > - correct me if I am wrong, but $\epsilon=0$ means $k = 1$ (from l.183), or again $k'=n$, i.e., mom = mean and not mom = median
> >
> > - line 209, when the inequation on $C$ implies the inequation on $k$ and $k'$, it seems to me that the formulas without the floor have been used, and that considering the formulas of l. 206 would require some modifications
> >
> > - I indeed feel that optimizing w.r.t. $\varepsilon$ would help the understanding (in the present case it is hard to analyze the limit scenarios)
> >
> > - mom or MoM does not make such a great difference (gd would be understood as "gradient descent" even in lower case), I really suggest to change in order to avoid confusion
> >
> > - some other typos: l.118-122 and l.218 $\epsilon$ is used to refer to different quantities, l.109-112 the action set is $\cal{X}$, or $X$, or $A$

---

> > > ### Author Response · Authors · 2021-09-02
> > > **Response**
> > >
> > > Thanks for your reply.
> > >
> > > 1. We will clarify the motivation of using the mean-of-medians estimator instead of the median estimator in the revision. Thanks for your advice on comparing our Mean-of-Medians estimator with the median estimator. We add the median estimator to the experiment of Figure 2(a) for further comparison. The algorithm by median estimator can be viewed as a special case of our Mean-of-Medians estimator where $\varepsilon=1$. We share the results in the link below. The plot shows the average regret of 10 independent paths of algorithm ''SupBTC'' and ''SupBTC\_mom'', where the median estimator is added by setting $\varepsilon=1$. Each path contains 100000 iterations under Student t’s noise with df=3. We conclude that the median estimator behaves worse than our Mean-of-Median estimator. We will add more details in the next revision.
> > >
> > > 2. Thanks for your reporting the error of missing a file. The file ‘./data/mom_10000_student_3_noise.txt’ can be generated by the program, which we cut out for reducing file size. In fact, the error occurs when there is no folder named ''data'' in the present working directory. Please try again after creating the folder ''data'' manually. In addition, for quickly reproducing our paper, we share the data used for drawing Figures 1 and 2 in the following link.
> > >
> > > 3. The symmetry assumption is necessary here. We need this assumption is to ensure that the mean of the final estimator $X_{mom}$ is zero (see Theorem 3.2). Meanwhile, as claimed in our paper and previous feedback, this assumption corresponds to the previous standard assumption that the noise is mean-zero, which is also necessary for the standard bandit problem.
> > > 4. Yes, you are right. When $\varepsilon = 0$, the mean-of-medians estimator reduces to the mean estimator. However, the mean estimator is not valid even for the previous heavy-tailed problem (see e.g. (Bubeck et al., 2013)). In our paper, when $\varepsilon = 0$, we cannot find a sufficiently large $C$ satisfies the inequality in Line 209 and further obtain that $2k’e^{-k/8} \leq \delta/2$, which also indicates that the mean estimator fails to tackle the super heavy-tailed bandit problems. This is why we restrict $\varepsilon \in (0, 1)$. We will add more comments on why we cannot simply use mean or median in the next revision.
> > >
> > > 5. Obtaining the optimal $\varepsilon$ is hard for us. But we can obtain the approximate solution. Specifically, the optimal $\varepsilon$ minimizes $\tilde{n}$ in Theorem 4.1. Then we can simply let $\big(16\log(2T/\delta)\bigr)^{1/\varepsilon} =  (4^{2/\alpha}\log(4/\delta))^{\frac{1}{1-\varepsilon}}$ and obtain the solution $\varepsilon \approx \frac{\log\log(T/\delta)}{\log\log(T/\delta) + \log\log(1/\delta) + (2/\alpha) \cdot \log4}$. More numerical verification is also provided in the link below. If a more precise solution is needed, you can use the binary search method. We also want to highlight that our point is not optimizing the $\varepsilon$, but studying the super heavy-tailed bandit problems and obtaining a near-optimal regret in $T$.
> > >
> > > 6. In Line 209, the reason why the inequality on $C$ implies the inequality on $k$ and $k’$ is that the function $2C^{1 - \varepsilon}e^{-C^{\varepsilon}/16}$ is monotone (for sufficiently large number). We believe we do not need the floor here. Thanks for mentioning this issue and we will double-check our whole proof carefully.
> > >
> > > 7. We will change the abbreviation and fix other typos.
> > >
> > > Link: https://ufile.io/56ti0ka1
> > >
> > > Thanks for your advice. If you still have some concerns, we are happy to discuss them.

---

> > > > ### Comment · Reviewer_4arc · 2021-09-02
> > > > **Response**
> > > >
> > > > Thank you for your detailed feedback. I think the median benchmark should be added to all plots if possible. Please find below some technical remarks.
> > > >
> > > > 3. I might miss something here, but it seems to me that it is sufficient for the r.v. $X$ to have an expected median (i.e., a 1/2 quantile) of 0, which is in particular verified if $X$ is symmetric. By linearity of the expectation, don't we simply get $\mathbb{E}[X_{mom}] = q_{1/2}(X)$? This seems to me the true analog "we use the mean (or the median-of-means), we want $X$ to be mean-0", "we use the median (or the mean-of-medians), we want $X$ to be median-0". Though, agreed symmetric is enough and an acceptable assumption (if necessary).
> > > >
> > > > 6. The way I understand l.209 is as follows: $2k'e^{-k/8} = 2\frac{n}{k}e^{-k/8} = 2 \frac{n}{n^\varepsilon}e^{-n^\varepsilon/8} = 2 n^{1-\varepsilon}e^{-n^\varepsilon/8}$. So, if $n \ge max(n_0, C)$, where $n_0$ is the point where $C \mapsto C^{1-\varepsilon}e^{-C^\varepsilon/16}$ becomes decreasing, then $2k'e^{-k/8} \le e^{-k/16} \le \delta/2$. But in the first series of equations, the floors were omitted.
> > > >
> > > > But my main concern is about the role of $\varepsilon$:
> > > > - by changing $\varepsilon$, one changes the number of blocks. While for MoM there is a precise choice dictated by the theory, I am surprised it is not the case here. Would you have an explanation?
> > > > - naively, I would choose $\varepsilon$ to obtain the best rate, and this leads to $\varepsilon = 0$, or mom = mean, although I know the empirical mean is a bad estimator. Even if $\varepsilon = 0$ is impossible, I find this limit behavior surprising too, how could it be explained? And note that with $k = \lfloor n^\varepsilon\rfloor$, there is an $\varepsilon > 0$ such that $k = 1$, so one should exclude more than just the point 0 (or consider ceiling instead of flooring for $k$)
> > > > - the confusion might come from $C$ (which is actually a function of $\varepsilon$ and I think should be denoted $C(\varepsilon)$ to make this explicit) which explodes as $\varepsilon$ goes to zero. So there seems to be some kind a tradeoff, which seems necessary to solve, or at least to mention. This parameter being the only one in the approach, I definitely feel that a detailed discussion is needed
> > > >
> > > > I have raised my score to 5, as an acknowledgement of the feedback provided by the authors so far. I am ready to raise it to 6 if authors provide a convincing explanation to the $\varepsilon$ behavior described above, that I find quite counterintuitive.

---

> > > > > ### Author Response · Authors · 2021-09-05
> > > > > **Response**
> > > > >
> > > > > Thanks for your re-evaluation and advice.
> > > > >
> > > > > - We believe it is insufficient to only assume r.v. $X$ to have a median of 0. This is because the empirical median may not be a good estimator of the true median. Consider the following case: suppose $\lbrace X_i\rbrace_{i = 1, 2}$ take values in $\lbrace -4, -2, 2, 6\rbrace$ and  $\Pr(X_i = -4) = \Pr(X_i = -2) = \Pr(X_i = 2) = \Pr(X_i = 6) = 1/4$. Then the median of $X_i$ is 0. However, $\mathbb{E} [\text{median}(X_1, X_2)] = 1/16 \cdot \sum_{x_1, x_2 \in \lbrace -4, -2, 2, 6\rbrace}(x_1+x_2)/2 = 1/16 \cdot (-4 -3 -1 + 1 -3 -2 + 0 + 2 -1 + 0 + 2 + 3 + 1 + 2 +4 + 6) = 7/16 \neq 0$.
> > > > > - Yes, you are right. We will fix this issue in the next revision.
> > > > >
> > > > > Then we make some comments on the role of $\varepsilon$ to address your main concerns.
> > > > >
> > > > > - Let us consider the sample complexity. Solving the inequality $|X_{mom}| \le \zeta$ in Line 203 gives that $\tilde{n} \ge \bigl(\frac{4^{2/\alpha}}{\zeta^2} \cdot \log(2/\delta)\bigr)^{\frac{1}{1 - \varepsilon}}$. Together with the constraint that $\tilde{n} \ge \max\lbrace C, (16\log(1/\delta))^{1/\varepsilon}\rbrace$ in Line 202, we have $\tilde{n} \ge \max\lbrace C, (16\log(1/\delta))^{1/\varepsilon}, \bigl(\frac{4^{2/\alpha}}{\zeta^2} \cdot \log(2/\delta)\bigr)^{\frac{1}{1 - \varepsilon}}\rbrace$. If we choose $\varepsilon$ near to $0$, $C$ and  $(16\log(1/\delta))^{1/\varepsilon}$ are large. If we choose $\varepsilon$ near to $1$, $\bigl(\frac{4^{2/\alpha}}{\zeta^2} \cdot \log(2/\delta)\bigr)^{\frac{1}{1 - \varepsilon}}$ is large. In other words, there is a tradeoff in $\varepsilon$ to balance these three terms.
> > > > > - Due to the above tradeoff, we cannot solve the optimal $\varepsilon$ explicitly. However, we have provided several methods to choose $\varepsilon$. (i) In our paper, we can simply choose $\varepsilon = 1/2$. With this simple choice, our algorithm also outperforms other methods. (ii) Theoretically, we can obtain the approximate solution of $\varepsilon$. We also conduct a numerical experiment to verify our theory. See our previous response for details. (iii) We can obtain the optimal $\varepsilon$ by some other algorithms, such as binary search.
> > > > > - We also want to highlight that $\bigl(16 \log(2T/\delta)\bigr)^{1/\varepsilon}$ (in Theorem 4.1) is the dominant term when $T$ is sufficient large, which indicates that $\tilde{n}$ scales as $polylog(T)$ for ***any*** fixed $\varepsilon \in (0, 1)$. Thus, our regret is optimal up to some logarithmic factors.
> > > > >
> > > > > Then we answer your questions.
> > > > >
> > > > > - Our analysis is different from MoM because we use an additional truncation technique. Hence, we need to tackle the tail probability $2k’ e^{k/8}$ in Line 207-208. To make this tail probability small, we choose $k = \lfloor n^{\varepsilon} \rfloor$ for some $\varepsilon \in (0, 1)$. Previous work in MoM tune the $k$ directly and thus do not need to optimize the additional parameter $\varepsilon$.
> > > > > - By the above argument, the extreme values that are close to 0 (or 1) cause the exploded sample complexity.
> > > > > - Although we have mentioned that $C$ depends on $\varepsilon$ in the paper (such as Theorem 4.1), we will use the notation $C(\varepsilon)$ in the next revision. Thanks for your advice. For the tradeoff, please see the discussion above.
> > > > > - We will incorporate your other suggestions in the next revision of our paper.
> > > > >
> > > > >
> > > > > We hope we have addressed your main concerns. If you have other concerns, we are happy to discuss them.

---

> > > > > > ### Comment · Reviewer_4arc · 2021-09-05
> > > > > > **Response**
> > > > > >
> > > > > > Thanks for your answer.
> > > > > >
> > > > > > - It seems to me that you designed a distribution whose $1/2$-quantile is the entire interval $(-2, 2)$, and $7/16$ is indeed in this interval. I believe that $q_{1/2}(X) = ${$0$} is enough, as the sample median is an unbiased estimate of the distribution median.
> > > > > >
> > > > > > - I find this discussion in terms of sample complexity indeed clearer than focusing on the rate, the tradeoff appears clearly. Maybe the comparison to [Bubeck et al. 2013] could also be expressed in terms of sample complexities and not rates?
> > > > > >
> > > > > > In light of this interaction with the authors I raise my score to 6, and encourage them to implement the modifications we discussed, that I hope will improve the paper.

---

> > > > > > > ### Author Response · Authors · 2021-09-05
> > > > > > > **Response**
> > > > > > >
> > > > > > > Thanks for your patience and valuable advice during the discussion. We will incorporate your suggestions in the next revision of our paper.

---

> ### Author Response · Authors · 2021-08-22
> **HAS OUR RESPONSE ADDRESSED YOUR CONCERNS?**
>
> Hello reviewer 4arc, we would be grateful if you can confirm whether our response has addressed your concerns and let us know if any issues remain. To recap our response,
>
> • By a simple counterexample, we clarify why the simple median cannot solve our problem.
>
> • We clarify that our new estimator is definitely not due to the symmetry property. Even with the symmetry property, previous work cannot tackle the super heavy-tailed (bandit) problems.
>
> • We clarify the relationship of $\alpha$, $\varepsilon$, and $C$.
>
> Thank you.

---

### Decision · Program_Chairs · 2021-09-27

**Decision:**

Accept (Poster)

**Comment:**

Dear authors,

There was an intense discussion and still some disagreement between the reviewers.

Hence I decided to take a look at the paper.
The article requires a serious polishing as there are a lot of typos (both in text and math).
The considered idea is interesting.

When applied to the bandit scenario, one difficulty when working with means of medians instead of means is the possible inversion of the order of arms that is, we may solve the wrong problem, since no longer targeting an arm with highest mean (when this exists).
Now, I appreciate the authors mention the  symmetry assumption, as it seems indeed crucial to avoid such an issue.
Actually, I realize that the proof of Theorem 4.1 does not mention this important fact: When doing the reduction, it is not enough that the pb reduces to the regret of a algorithm facing sub-Gaussian arms: the order of the arms should also be the same.
Luckily, this seems to be true. There seems to be also an important missing part on l.252, to go from applying to Theorem 3.2 to some property of the noise. I believe the statement should be about the mean of medians first.
[Note that on l.151, when you say that for alpha>1, we can relax the symmetry assumption to only considering centered noise, I am not sure this is correct, and suggest you remove this claim.]

The final result shows a reduction, that, in principle could potentially be arbitrarily bad (if $\\tilde n$ is too close to $T$ for instance).
Now, Corollary 4.5 shows that in some situations the final regret is controlled.  What can be criticised is that the choice of $\\tilde n$ should depend on $T$ hence making the algorithm no longer anytime.

All in all, despite these points, I believe the authors make a descent work and that the paper can be accepted.
However, I strongly encourage the authors to polish the wording and maths of the article.